mechanical engineering

compliant electrodes, self-healing, CNT/PVA hydrogel

**Authors for correspondence:**
Bo Qian
e-mail: qianbo@ecust.edu.cn
Jin Li
e-mail: lijinme@ecust.edu.cn

# Dielectric elastomer actuators based on stretchable and self-healable hydrogel electrodes

Yang Gao, Xiaoliang Fang, Danhquang Tran, Kuan Ju, Bo Qian and Jin Li

School of Mechanical and Power Engineering, East China University of Science and Technology, Shanghai 200237, People's Republic of China

(iD) JL, 0000-0002-4400-747X

Dielectric elastomer actuator (DEA) based on dielectric elastomer holds promising applications in soft robotics. Compliant electrodes with large stretchability and high electrical conductivity are the vital components for the DEAs. In this study, a type of DEA was developed using carbon nanotube/polyvinyl alcohol (CNT/PVA) hydrogel electrodes. The CNT/PVA hydrogel electrodes demonstrate a stretchability up to 200% with a small relative resistance change of approximately 1.2, and a self-healing capability. The areal strain of the DEA based on the CNT/PVA hydrogel electrodes is more than 40%, much higher than the ones based on pure PVA electrodes.

## 1. Introduction

Conventionally, rigid-bodied robots have made great progress in the field of automatically assembling and manufacturing [1–3]. The rigid actuations and mechanisms have limited compliance, leading to the potential threats of conventional robots for interaction with humans [1]. In contrast to rigid-bodied counterparts, soft robots composed of intrinsically soft and/or extensible materials with relatively large number of freedom degrees provide an opportunity to bridge the gap between machines and human being, since they can deform and absorb much of the energy arising from a collision [1,4–7].

Development of soft bodies with large deformation, high energy density and short response time is key challenge to fabricate soft robots [5]. So far, pneumatic-, electric-, chemical- and photonic-actuated soft robots have been extensively studied [1]. Dielectric elastomer actuator (DEA) among them, composed of a soft dielectric polymer sandwiched between two compliant electrodes, has drawn considerable attention for actuating soft

robots efficiently using electrical potential [4,8–12]. When they are subjected to a voltage, the opposite electric charges accumulated on the faces of the polymer lead to the reduction in the thickness of the polymer and thus the expansion of its area [3]. Compliant electrode is an important component for DEAs, since they must be able to synchronously follow large strains of the elastomer without generating an opposing stress or losing conductivity [13]. Therefore, properties such as large stretchability and high electrical conductivity are highly demanded for compliant electrodes [14–23]. A variety of materials including carbon grease [13], carbon nanotubes (CNTs) [15], nanowires [14] and graphene [16] have been investigated as the electrodes for DEAs up to now. Carbon grease is widely used [17,24], a highest areal strain of 1692% has been reported by Suo's group [25,26]. CNTs [15], graphite powder [13] and corrugated metal [21,22] directly deposited on the dielectric polymer are able to generate large actuation strains, but it is found that these materials have poor mechanical adhesion with DEAs, and are not applicable for miniaturization [14,27]. Stretchable composite materials prepared by incorporation of CNTs, nanowires or graphene into elastomer are alternative electrodes for the DEAs. Electrical conductivity of the composite electrodes is typically low and its stiffness is generally higher than dielectric polymers [17,28]. Furthermore, there is a trade-off between electrical conductivity and stiffness of the composite electrodes: to increase electrical conductivity of the composite electrodes by filling more active material will increase their stiffness [22,29]. Ionic conductive polymers have been investigated as potential candidates to compliant electrodes for DEAs due to their high electrical conductivity at large deformation [17–19]. Suo's group developed a compliant electrode based on polyacrylamide hydrogel, which can reach an areal strain up to 134% at an electrical field of 124 MV m$^{-1}$ [18].

In this study, a highly stretchable and self-healable ionic conductive electrode was developed as a compliant electrode for DEAs. The electrode was prepared by incorporation of CNTs into polyvinyl alcohol (PVA) hydrogel [23]. The CNT/PVA hydrogel electrode is highly stretchable and conductive, with a relative resistance change of 1.2 at a strain up to 200%. Moreover, different from the ionic conductive electrodes reported for DEAs so far [17–19], the CNT/PVA composite electrode is self-healable due to the hydrogen-bonding between tetrafunctional borate ion and –OH group. Areal strain generated by DEAs based on CNT/PVA hydrogel electrode are almost two times larger than that based on pure PVA hydrogel electrode. Large stretchability, high electrical conductivity and self-healing capability of the CNT/PVA composite electrode prospects its applications in DEAs.

# 2. Experimental

## 2.1. Preparation of CNT/PVA hydrogel electrodes

An amount of 0.32 g PVA (Mw = ~145 000, SIGMA-ALDRICH, Co., Germany) with 3 ml deionized (DI) water was heated in water bath at 90°C until PVA was uniformly dissolved. Second, 50 mg multi-wall CNTs (Shengzhen Nanotech Port Co. Ltd) with an average diameter approximately 30 nm was dispersed in 5 ml DI water with sodium dodecyl benzene sulfonate (SDBS, Aladdin Industrial Co., Shanghai, China) as a surfactant and ultrasonicated for approximately 2 h. After that, the two above aqueous solutions were mixed together under magnetic stirring for approximately 1 h. Then, sodium tetraborate with the concentration of 40 mg ml$^{-1}$ was added into the mixed CNT/PVA solution to form the CNT/PVA hydrogel.

## 2.2. Fabrication of DEAs based on CNT/PVA hydrogel electrodes

The DEAs have a sandwiched structure. Having a layer of dielectric elastomer VHB 4910 (3M Company Shanghai Branch) fixed on a rigid acrylic frame with radius $D$, two CNT/PVA hydrogel films with radius $d$ ($d < D$) were attached to the top and bottom faces of the dielectric elastomer membrane, respectively. Two pieces of copper tape aiming at electromechanical measurements were bonded to the ends of the electrodes afterwards.

## 2.3. Characterizations of DEAs

The electromechanical performance of CNT/PVA hydrogel electrodes was recorded using an electrochemical workstation (CHI660E, Shanghai Chenhua Inc.). The relative resistance change ($S$) of

CNT/PVA hydrogel electrodes was defined as follows:

$$S = \frac{(R - R_o)/R_o}{\varepsilon},$$ (2.1)

where $R_o$ is the initial resistance of the CNT/PVA electrode at $\varepsilon = 0\%$; $\varepsilon$ and $R$ are the uniaxial strain applied to the electrodes and the corresponding resistance, respectively.

A voltage signal was amplified by a high-voltage amplifier (TRC-2020P, Teslaman) to actuate the DEAs. A thin paper tape was attached on the edge of electrode area of the DEAs for measuring real-time in-plane displacement. A laser displacement sensor (LK-G4000A, Kenyence) was used for tracking real-time displacement of the tape continuously. Areal strain ($\varepsilon_{\text{area}}$) of the DEAs was calculated as follows:

$$\varepsilon_{\text{area}} = \frac{\pi d'^2 - \pi d^2}{\pi d^2} \times 100\%,$$ (2.2)

where $d$ and $d'$ are the radii of the active region sandwiched by the electrodes at original and actuated state, respectively.

# 3. Results and discussion

The fabrication procedure of the DEAs based on CNT/PVA hydrogel electrodes is shown in figure 1. Figure 1a shows the schematic illustrating the preparation of CNT/PVA hydrogel electrodes [23]. PVA aqueous solution and CNT solution were mixed together to form the CNT/PVA hydrogel (figure 1b) by introducing sodium tetraborate solution. Infilling of CNTs seen from electronic supplementary material, figure S1 and S2 improves electrical conductivity of PVA up to approximately 0.71 S cm$^{-1}$ higher than that of pure PVA (approx. 0.22 S cm$^{-1}$), while the CNTs seem to be wrapped by PVA resulting from high amount of PVA. Figure 1c schematically demonstrates a circular DEA with a dielectric elastomer (VHB 4910) sandwiched by two CNT/PVA hydrogel electrodes. The VHB elastomer was pre-stretched to a strain 200% × 200% in two directions and fixed to a rigid acrylic frame. Then two pieces of CNT/PVA hydrogel were applied to the surfaces of the elastomer serving as the top and bottom electrodes, respectively. Figure 1d shows a photography of a DEA based on CNT/PVA hydrogel electrode. Figure 1e shows the active region of the DEA, with a diameter of approximately 3 cm. Uniaxial tensile tests were conducted to obtain the mechanical performance of the hydrogel electrode and VHB dielectric layer, as shown in electronic supplementary material, figure S3. The CNT/PVA hydrogel electrodes have high failure strain close to 780% with fracture strength of approximately 12 kPa, while VHB layer has failure strain of approximately 930% and fracture strength of approximately 450 kPa. In addition, stress of hydrogel electrodes at all strains is less than the VHB dielectric layer, ensuring hydrogel electrodes are compliant to the dielectric layer.

Before the study of the electromechanical properties of the DEA mentioned above, the electrical response behaviour of the CNT/PVA hydrogel electrodes subjected to strains was systematically investigated and is shown in figure 2. Figure 2a shows the schematic of a CNT/PVA hydrogel electrode used for the electromechanical measurement. A thin sheet of CNT/PVA hydrogel with a size of 30 × 10 mm was sandwiched between two layers of VHB 4910. Two copper tapes were bonded at the two ends of the CNT/PVA sheet to work as the current collectors during electromechanical measurements. Figure 2b shows the current–voltage curves of the CNT/PVA hydrogel with 50 mg CNT at different strain. As the applied strain increases, the slope of the current–voltage curves decrease indicating the increase in the resistance of the CNT/PVA hydrogel electrodes. High conductivity with large stretchability and good linearity of the electrodes are necessary in the field of DEAs. Hydrogels with different amount of CNTs are investigated to obtain the optimized electrode for the DEA. Electronic supplementary material, figure S4 shows the relative resistance changes of hydrogels with different amount of CNTs. The sample with 25 mg CNT is slightly higher than that of the hydrogel with 50 mg CNT at higher strain (greater than approx. 100%). Although the hydrogel with 75 mg CNT has better electrical conductivity than the one with 50 mg CNT, it cannot reach strain over approximately 150%. Therefore, hydrogel with 50 mg CNT was chosen as the electrode for the following study. As shown in figure 2c, the electrical curve of the CNT/PVA hydrogel electrode has a high linearity up to strain of 200% and much lower relative resistance change of approximately 1.2 than other stretchable conductors [29–31], demonstrating the potential applications in DEAs. Figure 2d shows the hysteresis curves of the CNT/PVA hydrogel electrode under different strains of 25%, 50%,

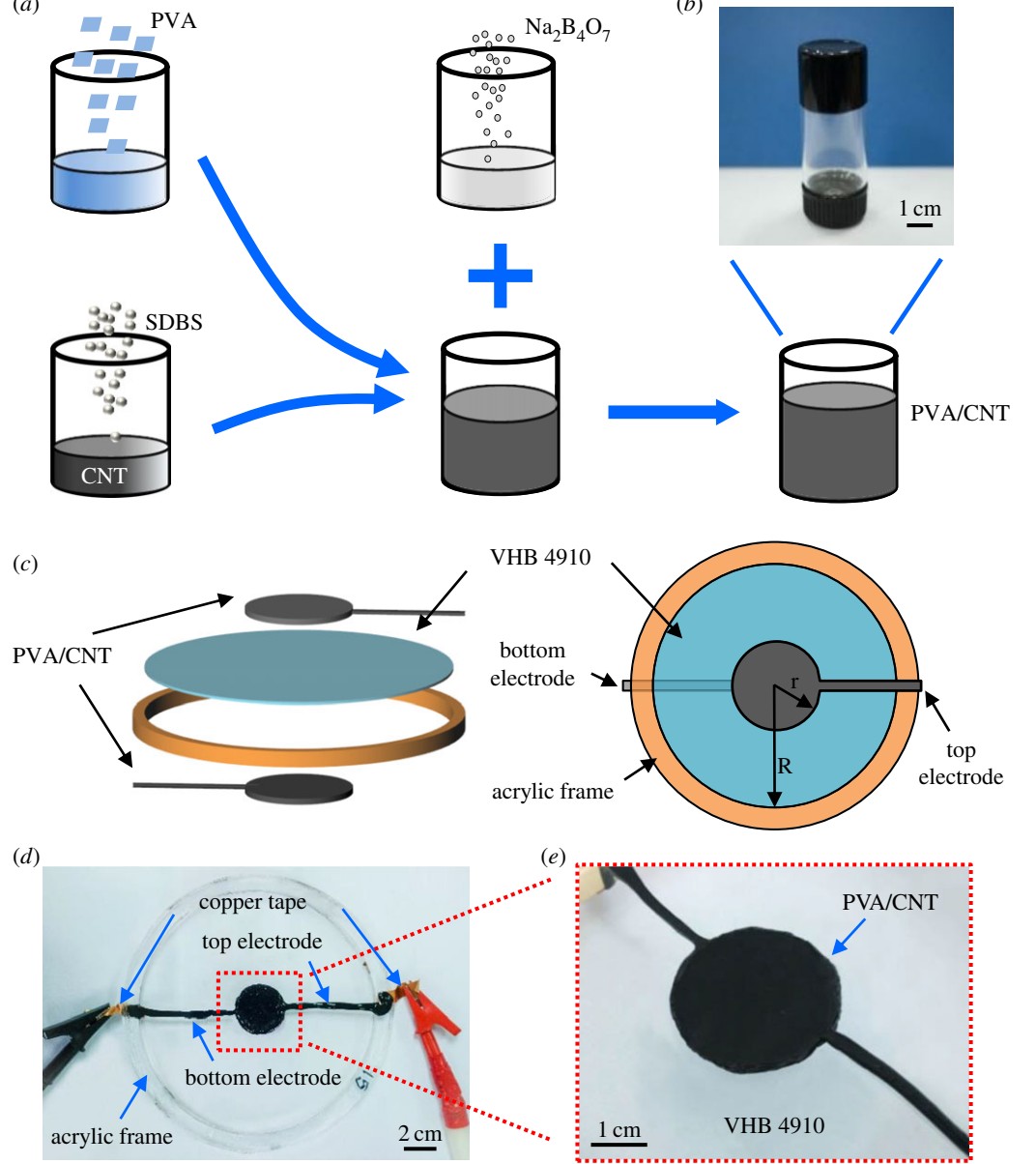

**Figure 1.** (*a*) A schematic of preparation of CNT/PVA hydrogel electrodes. (*b*) A photograph of CNT/PVA hydrogel. (*c*) A schematic demonstrating the assembly of a circular DEA composed of CNT/PVA hydrogel electrode and VHB dielectric. (*d*) A photograph of circular DEA based on CNT/PVA hydrogel electrode. (*e*) A photograph showing the active area of the DEA.

100%, 150% and 200%, respectively. During the stretching–releasing test, the signal remained stable without any distinct drift. The existing hysteresis loops can be attributed to the hysteresis of the VHB elastomer or the PVA used for hydrogel preparation. Durability is another significant characteristic that determines the lifetime of a compliant electrode. Hence, a stretching–releasing cyclic test was performed to investigate the durability of the CNT/PVA hydrogel under a strain of 20%, as shown in figure 2*e*. The hydrogel can maintain its performance up to approximately 1800 cycles. After that, the relative resistance change increased by approximately 20% and then became stable for another approximately 1700 cycles. The performance degradation may be due to the creep property of the VHB tape for long cycles, which is still under investigation.

Electronic supplementary material, figures S5 and S3 demonstrate the self-healing capability of CNT/PVA hydrogel electrodes. Electronic supplementary material, figure S5 shows a stretching testing for the electrode after self-healing. CNT/PVA hydrogel was firstly clamped by two fixtures (shown in electronic supplementary material, figure S5a), then the hydrogel strip was cut off with scissors and it was separated into two individual parts, as shown in electronic supplementary material, figure S5b. After

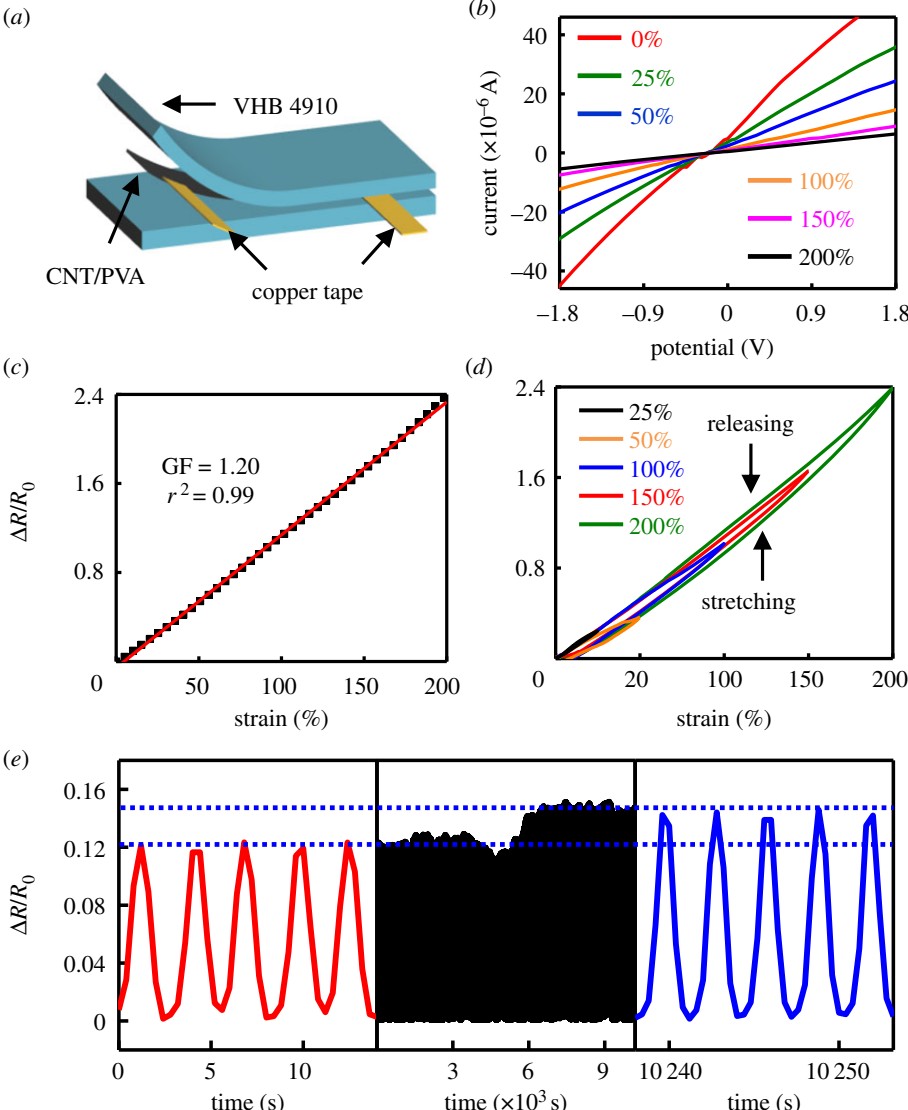

**Figure 2.** Electromechanical performance of the CNT/PVA hydrogel electrodes. (*a*) A schematic showing the device structure of CNT/PVA hydrogel electrodes for electromechanical testing. (*b*) Current–voltage curves of the electrode at different applied strains. (*c*) Relative resistance changes of the electrode plotted as a function of applied strains. (*d*) Hysteresis performance of the electrode with applied strains ranging from 25% to 200%. (*e*) Durability test of the electrode under 20% strain.

that, the two parts were contacted again for approximately 1 min for its self-healing (electronic supplementary material, figure S5c). The CNT/PVA hydrogel was stretched to the strain of approximately 100% (electronic supplementary material, figure S5d) and approximately 170% (electronic supplementary material, figure S5e) without having obvious structural fracture, indicating that CNT/PVA hydrogel electrode still keeps the large stretchability after self-healing. A visualized experiment was further conducted to demonstrate the self-healing properties of the CNT/PVA hydrogel electrodes (figure 3a–c). An LED was connected to a power source using the CNT/PVA hydrogel electrodes. The LED was successfully lit at a driving voltage of 1 V (figure 3a). When one the CNT/PVA electrode was completely cut off, the LED was extinguished due to the open-circuit state (figure 3b). The LED was lit again (figure 3c) when the two furcate sections contacted each other without any other external stimuli, such as force, heat and light [7]. The self-healing characteristic of the CNT/PVA electrodes is ascribed to the hydrogen-bonding between tetrafunctional borate ion and –OH group [23,32]. Figure 3d shows the resistance change recorded by an electrochemical workstation during cutting and healing processes in the visualized experiment. The resistance rises up quickly and an open-circuit state was reached once the CNT/PVA was cut off (figure 3d); and the resistance drops rapidly and returns to the original conductive state as the two fractured sections contact each other

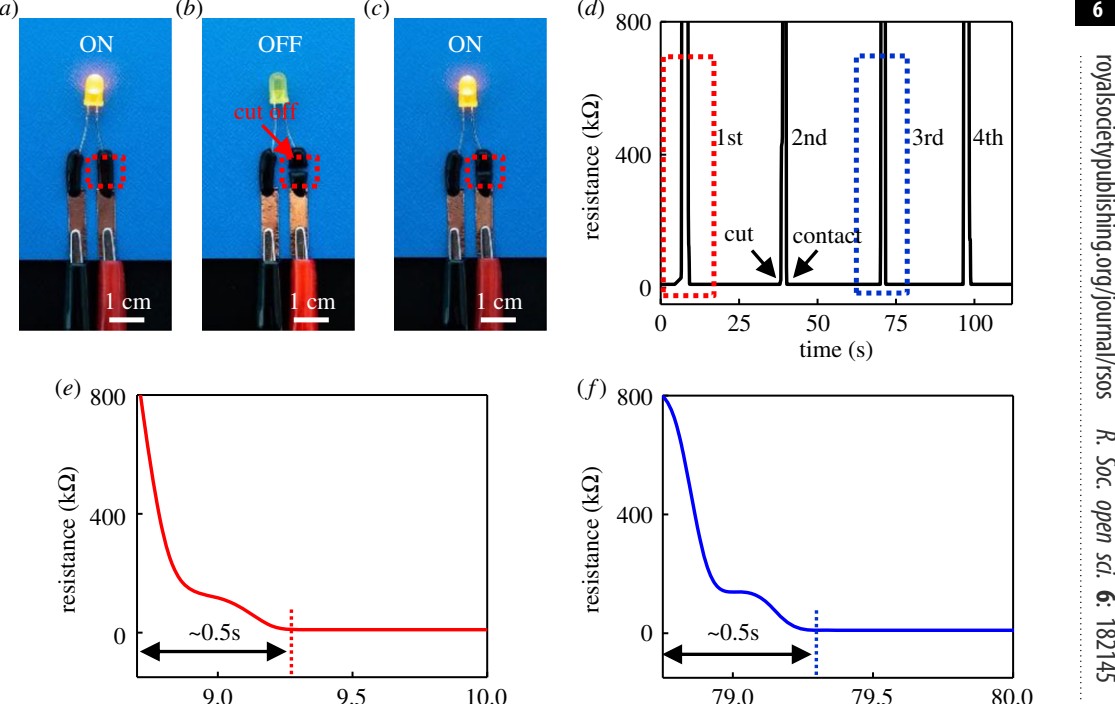

**Figure 3.** (*a*–*c*) A visualized experiment showing the self-healing capability of the CNT/PVA hydrogel electrodes. (*d*) Cycling of cutting/healing process of the CNT/PVA hydrogel electrodes. (*e*,*f*) The response time for the healing process of the CNT/PVA hydrogel electrodes.

within approximately 0.5 s (figure 3*e*,*f*). The cutting–healing experiment was repeated for four times (figure 3*d*), revealing a stable self-healing property of the CNT/PVA hydrogel electrodes. Stability of the hydrogel after the cutting–healing test was further investigated by subjecting it to a cycling test for 2000 cycles (electronic supplementary material, figure S6). The hydrogel electrode can maintain its electromechanical performance within approximately 500 cycles. Then the electrical signal starts drifting. After 2000 cycles, the relative resistance change is two times larger than the original value. The possible reason is still under investigation.

Large stretchability, high conductivity and self-healing capability of the CNT/PVA hydrogel electrodes enable their potential application in DEAs. Figure 4 shows a systematic study of the electromechanical performance of the DEAs based on the CNT/PVA hydrogel electrodes. Areal strain of the DEAs was measured using the experimental set-up shown in figure 4*a*. Detailed information can be found in the Experimental section. Thickness of the electrode in the experiment is approximately 1 mm. Figure 4*b* and electronic supplementary material, Movie M1 show the deformation of a DEA under electrical stimulation. The DEA was coated with the CNT/PVA hydrogel electrodes on both surfaces of the dielectric elastomer to form a circle active region. When the DEA is subjected to a high voltage, it reduces in thickness and expands in area due to the effect of Maxwell stress [3]. Actuation performance of the DEA based on the CNT/PVA hydrogel electrodes at different applied voltages is shown in figure 4*c*. With the increase of applied voltage, areal strain of the DEA increases. At an applied voltage of approximately 2 kV, the areal strain is more than 40%. Compared with DEAs based on pure PVA hydrogel, DEAs using the CNT/PVA hydrogel electrodes are much enhanced. Areal strain of the DEA on VHB 4905 (blue curve in figure 4*c*) is higher than that based on VHB 4910 (black curve in figure 4*c*), while the former breaks down at 1.8 kV. The lower breakdown voltage of the DEA based on VHB 4905 can be attributed to lower thickness during expansion. The actuation properties of the DEAs based on different electrodes was measured at different frequencies under approximately 1.5 kV, as shown in figure 4*d*. Areal strain of both the two samples reduced as the frequency of applied voltage increased. DEAs based on the CNT/PVA hydrogel electrodes provide larger areal strains at all the frequencies than the ones based on pure PVA hydrogel, suggesting a better frequency response behaviour of the actuator enabled by the CNT/PVA hydrogel electrodes. The DEA based on CNT/PVA hydrogel with VHB 4905 (blue curve in figure 4*d*) has two times larger areal strain than that of the device based on VHB 4910 (blue curve in figure 4*d*).

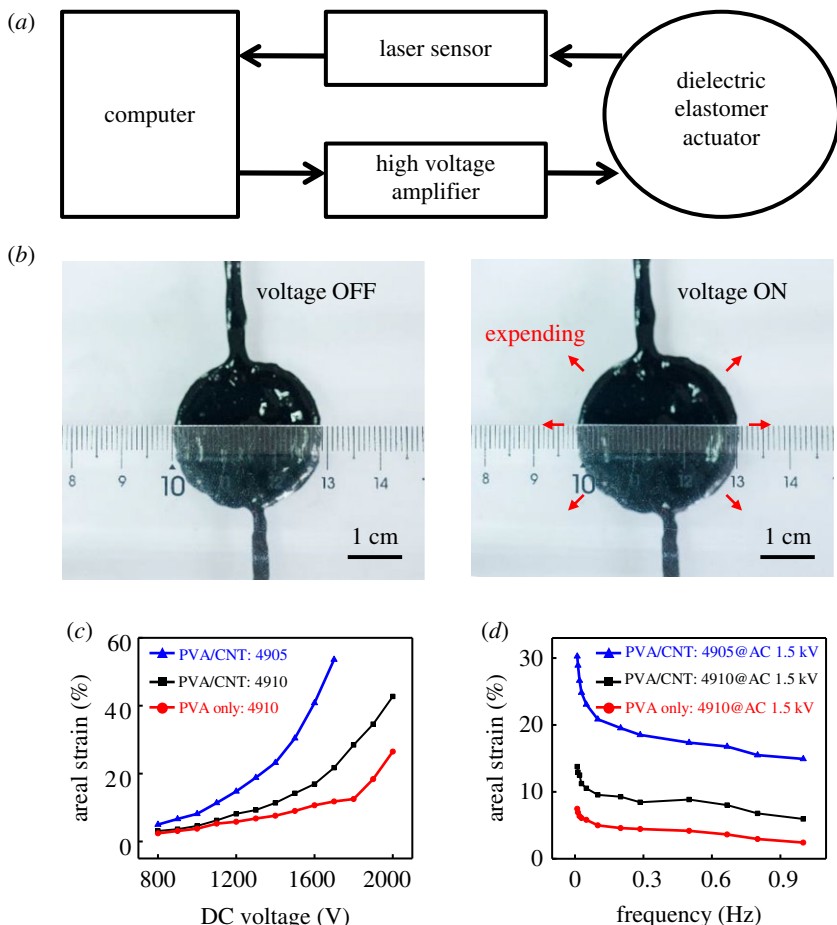

**Figure 4.** (a) Block diagram of strain measurement experiment set-up for the DEAs based on CNT/PVA hydrogel electrodes. (b) Voltage-induced deformation in the active region of the DEA based on CNT/PVA hydrogel electrodes. (c) Areal strain of the DEA based on CNT/PVA hydrogels with VHB 4905 and 4910, and PVA hydrogel electrodes with VHB 4910 as a function of applied voltage, respectively. (d) Areal strain of the DEA based on CNT/PVA hydrogels with VHB 4905 and 4910, and PVA hydrogel electrodes with VHB 4910 as a function of applied frequency under AC 1.5 kV, respectively.

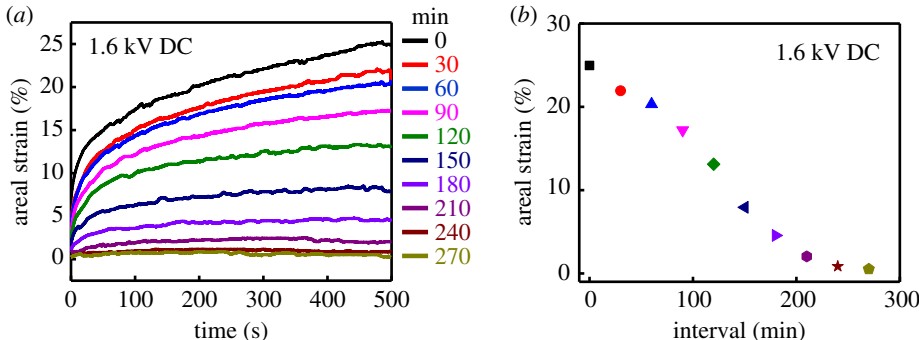

**Figure 5.** (a) The areal strains of CNT/PVA hydrogel during the loss of water. (b) The degradation of the areal strain of the DEA plotted as a function of time.

Although the CNT/PVA hydrogel electrodes not only provide high electromechanical properties but also have the capability of self-healing, it is notable that electrical performance of the electrode can become degraded due to the loss of water. Effect of water loss on the actuation property of the DEAs based on CNT/PVA hydrogel electrodes was studied, as shown in figure 5. In the experiment, the actuated areal strain of the DEA was measured once every 30 min. For each measurement, the applied voltage was maintained for 500 s. Figure 5a illustrates the time-dependence areal strains of the DEA when it is excited by a DC voltage of 1.6 kV. The results show that the deformation of the DEA creeps slowly and needs time to reach its equilibrium state. The areal strain of the DEA is affected by the

loss of water. As time goes by, the CNT/PVA hydrogel electrodes gradually lose their elasticity due to the evaporation of water in the electrode. As a consequence, the areal strain of the DEA decreases from 25% at 0 min to 1% at 270 min after about 4 h exposure to air (figure 5b). Packing with soft materials or using ionic liquid to replace the water could be the potential solution to address the challenge.

# 4. Conclusion

In this study, a type of DEAs was developed based on compliant and self-healable CNT/PVA hydrogel electrodes. The CNT/PVA hydrogel electrodes have a stretchability up to 200%, with a small relative resistance change of approximately 1.2. Due to the hydrogen bonding between tetrafunctional borate ion and –OH group, the CNT/PVA hydrogel electrodes have a self-healing capability. When the CNT/PVA hydrogel electrodes are attached to the surfaces of dielectric polymer to form the DEAs, a maximum areal strain of more than 40% is reached, much higher than the pure PVA hydrogel electrodes.

Data accessibility. Supporting data files, including SEM images, strain–stress curves and sequential pictures showing stretching test of the electrode, have been uploaded as part of the electronic supplementary material. Raw data tables of the electrode, including electrical performance files, duration test files and water loss files have been deposited to Dryad at: https://doi.org/10.5061/dryad.jd03548 [33].
Authors' contributions. Y.G. and X.F. contributed to the conception, the design of the experiment and drafting the manuscript. X.F., D.T. and K.J contributed to the conduction of experiments. B.Q. and J.L. contributed to the drafting of the manuscript.
Competing interests. We have no competing interests.
Funding. This project is supported by National Key Research and Development Program of China (grant no. 2018YFB1105400), National Natural Science Foundation of China (grant no. 51705154) and Shanghai Program for Professor of Special Appointment (Eastern Scholar) at Shanghai Institutions of Higher Learning. This work is also financially sponsored by Natural Science Foundation of Shanghai (grant no. 19ZR1413300), Shanghai Rising-Star Program (A type) (grant no. 18QA1401300) and Shanghai Sailing Program 17YF1403300.
Acknowledgements. We thank our colleagues in Lab B11, whose thoughtful criticisms and encouragement spurred us on. Specially, X.F., one of the co-authors, thanks his girlfriend for her company during the repetitive experiments, and her infectious energy and enthusiasm.

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
