## [Reviewer comments · Royal Society Open Science]

Review History

RSOS-182145.R0 (Original submission)

Review form: Reviewer 1

Is the manuscript scientifically sound in its present form?

Yes

Are the interpretations and conclusions justified by the results?

Yes

Is the language acceptable?

Yes

Is it clear how to access all supporting data?

Yes

Do you have any ethical concerns with this paper?

No

Have you any concerns about statistical analyses in this paper?

No

Recommendation?

Accept with minor revision (please list in comments)

Comments to the Author(s)

The authors reported a type of dielectric elastomer actuator, which is using carbon nanotube/polyvinyl alcohol (CNT/PVA) hydrogel as the stretchable electrodes. The CNT/PVA hydrogel electrodes demonstrate a stretchability up to 200% with a small relative resistance change of ~ 1.2 , and a self-healing capability. Even though the actuated deformation strain is not so high, the healable electrode is quite interesting. I think this paper can fit the publication level of R. Soc. open sci. I only have few minor suggestions.

1) The electrode in figure 1e seems to be not flat. It is better to show the photos of some optimized sample with clear and flat electrode.

2) VHB 4910 is a rather thick tape and you can use some thinner ones (4905 or 9473) to realize high expansion strain.

3) Usually self-healing capability of DEA is related to the electrical breakdown effect, not just the cutting behavior. It is better you can discuss something about the healing behavior after electrical breakdown.

4) In figure 2c and 2d, the author discussed about resistance changing. It is better to show the conductivity changing rather than resistance. Then, you may find that the conductivity is almost unchanged with different strain, which would be more persuasive.

5) Recently, people can produce high voltage to drive DEA by using self-powered technique [Adv. Funct. Mater. 26, (2016) 4906-4913; Nano Energy 38 (2017) 91-100; Adv. Funct. Mater. (2017), 27, 1603788]. It is better the author can report some of these work in their introductions.

Review form: Reviewer 2

Is the manuscript scientifically sound in its present form?

Yes

Are the interpretations and conclusions justified by the results?

Yes

Is the language acceptable?

Yes

Is it clear how to access all supporting data?

Not Applicable

Do you have any ethical concerns with this paper?

No

Have you any concerns about statistical analyses in this paper?

No

Recommendation?

Accept with minor revision (please list in comments)

Comments to the Author(s)

The authors reported carbon nanotube and polyvinyl alcohol-based dielectric elastomer actuator that has 40% more areal strain and is 200% stretchable. The new structure also showed self-healing capability due to hydrogen bonding and lower relative resistance than the other stretchable conductors. I believe the authors have presented interesting work and have addressed the previous reviewers' questions. Therefore, I would recommend the manuscript for publication after a minor revision.

1. It is not elevated that why the authors have chosen specific percentages for the chemical mixes while preparing the CNT-PVA hydrogel. i.e., 0.32 g PVA, 50g CNT, 2 hrs ultrasonication. How did they decide these numbers and times? What happens otherwise?
2. It is stated that the hydrogel could repeat its performance up to 1800 cycles without any degradation. Did the authors repeat this lifetime test to the sample they cut after the self-healing process? Are there any different results observed?
3. Writing can be improved.

Decision letter (RSOS-182145.R0)

04-Jun-2019

Dear Dr Li

On behalf of the Editors, I am pleased to inform you that your Manuscript RSOS-182145 entitled "Dielectric Elastomer Actuators Based on Stretchable and Self-Healable Hydrogel Electrodes" has been accepted for publication in Royal Society Open Science subject to minor revision in accordance with the referee suggestions. Please find the referees' comments at the end of this email.

The reviewers and handling editors have recommended publication, but also suggest some minor revisions to your manuscript. Therefore, I invite you to respond to the comments and revise your manuscript.

- Ethics statement

- Data accessibility

If you wish to submit your supporting data or code to Dryad (<http://datadryad.org/>), or modify your current submission to dryad, please use the following link:
<http://datadryad.org/submit?journalID=RSOS&manu=RSOS-182145>

- **Competing interests**

- **Authors' contributions**

- **Acknowledgements**

- **Funding statement**

Because the schedule for publication is very tight, it is a condition of publication that you submit the revised version of your manuscript before 13-Jun-2019. Please note that the revision deadline will expire at 00.00am on this date. If you do not think you will be able to meet this date please let me know immediately.

When submitting your revised manuscript, you will be able to respond to the comments made by the referees and upload a file "Response to Referees" in "Section 6 - File Upload". You can use this to document any changes you make to the original manuscript. In order to expedite the

processing of the revised manuscript, please be as specific as possible in your response to the referees. We strongly recommend uploading two versions of your revised manuscript:

Kind regards,
Alice Power
Royal Society Open Science
openscience@royalsociety.org

on behalf of Professor Jun Fu (Associate Editor) and R. Kerry Rowe (Subject Editor)
openscience@royalsociety.org

Reviewer comments to Author:

Reviewer: 1

Comments to the Author(s)

The authors reported a type of dielectric elastomer actuator, which is using carbon nanotube/polyvinyl alcohol (CNT/PVA) hydrogel as the stretchable electrodes. The CNT/PVA hydrogel electrodes demonstrate a stretchability up to 200% with a small relative resistance change of ~ 1.2 , and a self-healing capability. Even though the actuated deformation strain is not so high, the healable electrode is quite interesting. I think this paper can fit the publication level of R. Soc. open sci. I only have few minor suggestions.

- 1) The electrode in figure 1e seems to be not flat. It is better to show the photos of some optimized sample with clear and flat electrode.
- 2) VHB 4910 is a rather thick tape and you can use some thinner ones (4905 or 9473) to realize high expansion strain.
- 3) Usually self-healing capability of DEA is related to the electrical breakdown effect, not just the cutting behavior. It is better you can discuss something about the healing behavior after electrical breakdown.
- 4) In figure 2c and 2d, the author discussed about resistance changing. It is better to show the conductivity changing rather than resistance. Then, you may find that the conductivity is almost unchanged with different strain, which would be more persuasive.
- 5) Recently, people can produce high voltage to drive DEA by using self-powered technique [Adv. Funct. Mater. 26, (2016) 4906-4913; Nano Energy 38 (2017) 91-100; Adv. Funct. Mater. (2017), 27, 1603788]. It is better the author can report some of these work in their introductions.

Reviewer: 2

Comments to the Author(s)

The authors reported carbon nanotube and polyvinyl alcohol-based dielectric elastomer actuator that has 40% more areal strain and is 200% stretchable. The new structure also showed self-healing capability due to hydrogen bonding and lower relative resistance than the other stretchable conductors. I believe the authors have presented interesting work and have addressed the previous reviewers' questions. Therefore, I would recommend the manuscript for publication after a minor revision.

1. It is not elevated that why the authors have chosen specific percentages for the chemical mixes while preparing the CNT-PVA hydrogel. i.e., 0.32 g PVA, 50g CNT, 2 hrs ultrasonication. How did they decide these numbers and times? What happens otherwise?
2. It is stated that the hydrogel could repeat its performance up to 1800 cycles without any degradation. Did the authors repeat this lifetime test to the sample they cut after the self-healing process? Are there any different results observed?
3. Writing can be improved.

Editorial Office Comments to Authors:

For more information about language-polishing services endorsed by the Royal Society, please follow the link below:

<https://royalsociety.org/journals/authors/language-polishing/>

Author's Response to Decision Letter for (RSOS-182145.R0)

See Appendix A.

Decision letter (RSOS-182145.R1)

11-Jul-2019

Dear Dr Li,

I am pleased to inform you that your manuscript entitled "Dielectric Elastomer Actuators Based on Stretchable and Self-Healable Hydrogel Electrodes" is now accepted for publication in Royal Society Open Science.

on behalf of Professor Jun Fu (Associate Editor) and R. Kerry Rowe (Subject Editor)
openscience@royalsociety.org

Appendix A

Yang Gao

Associate Professor

*School of Mechanical and Power Engineering,
East China University of Science and Technology*

+86-21-64253776

yanggao@ecust.edu.cn

June 14, 2019

Dear Dr. Alice Power:

Thank you for sending us the reviewer comments on our manuscript “*Dielectric Elastomer Actuators Based on Stretchable and Self-Healable Hydrogel Electrodes*” by Yang Gao, Xiaoliang Fang, Danhquang Tran, Kuan Ju, Bo Qian, Jin Li. We have fully addressed all of the inputs from the four reviewers. In the response letter below, we list their comments, our responses to them and our associated modifications to the manuscript. Finally, a list of changes is provided.

We feel that these modifications make the manuscript suitable for publication in *Royal Society Open Science*. This work represents a type of dielectric elastomer actuator (DEA) based on compliant and self-healable carbon nanotube/polyvinyl alcohol (CNT/PVA) hydrogel electrodes, which offers important contributions to the field of soft robotics.

We thank you in advance for your time and attention.

Sincerely,

Yang Gao
School of Mechanical and Power Engineering,
East China University of Science and Technology,
Shanghai 200237 (China)
E-mail: yanggao@ecust.edu.cn

Sincerely,
Jin Li

Reviewer #1

Comments:

The authors reported a type of dielectric elastomer actuator, which is using carbon nanotube/polyvinyl alcohol (CNT/PVA) hydrogel as the stretchable electrodes. The CNT/PVA hydrogel electrodes demonstrate a stretchability up to 200% with a small relative resistance change of ~1.2, and a self-healing capability. Even though the actuated deformation strain is not so high, the healable electrode is quite interesting. I think this paper can fit the publication level of R. Soc. open sci. I only have few minor suggestions.

Our response: We thank the Reviewer for the precious comments on our work.

Comment #1. The electrode in figure 1e seems to be not flat. It is better to show the photos of some optimized sample with clear and flat electrode.

Our response: We thank the Reviewer for the comments on our work. According to the suggestion from the Reviewer, we replace Figure 1e with a higher resolution to show the electrode.

Our modification to the manuscript:

(1) In the revised manuscript, Figure 1e is replaced by a higher resolution image, as shown in Figure R1.

Figure R1. A photograph of CNT/PVA hydrogel on DEA.

Comment #2. VHB 4910 is a rather thick tape and you can use some thinner ones (4905 or 9473) to realize high expansion strain.

Our response: We thank the Reviewer for the comments on our work.

(1) Expansion tests of DEAs based on VHB 4910 and VHB 4905 were conducted according to the suggestion from the Reviewer, as shown in Figure R2. The areal strain of the DEA based on VHB 4905, as shown in Figure R2a, is higher than that of

the DEA based on VHB 4910, while the former breakdown voltage is 1.8 kV. The lower breakdown voltage of DEA based on VHB 4905 can be contributed to its relative smaller thickness.

- (2) Furthermore, the DEA based on VHB 4905 was still tested likewise under 1.5 kV alternating current voltage with different frequency, as shown in Figure R2b. The DEA based on VHB 4905 has 2 times larger areal strain than that of the device based on VHB 4910.

Figure R2. (a) Areal strains of the DEAs based on CNT/PVA hydrogel electrodes with VHB 4905 and 4910, and PVA hydrogel electrodes with VHB 4910 as a function of applied voltage, respectively. (b) Areal strains of the DEAs based on CNT/PVA hydrogel electrodes with VHB 4905 and 4910, and PVA hydrogel electrodes with VHB 4910 as a function of applied frequency under AC 1.5 kV, respectively.

Our modification to the manuscript:

- (1) In the revised manuscript, Figure 4c and 4d and their figure captions are replaced.
- (2) In the revised manuscript, on page 7, line 13, “Areal strain of the DEA based on VHB 4905 (blue curve in Figure 4c) is higher than that based on VHB 4910 (black curve in Figure 4c), while the former breaks down at 1.8 kV. The lower breakdown voltage of the DEA based on VHB 4905 can be contributed to lower thickness during expansion.” is added.
- (3) In the revised manuscript, on page 7, line 21, “The DEA based on CNT/PVA hydrogel with VHB 4905 (blue curve in Figure 4d) has 2 times larger areal strain than that of the device based on VHB 4910 (blue curve in Figure 4d).” is added.

Comment #3. Usually self-healing capability of DEA is related to the electrical breakdown effect, not just the cutting behavior. It is better you can discuss something about the healing behavior after electrical breakdown.

Our response: We thank the Reviewer for the comments on our work.

- (1) While some groups have previously referred to electrical breakdown effect of DEAs in their works [*J. Polym. Sci. Pol. Phys.*, **2011**, 49, 504-515; *Int. J. Solids. Struct.*, **2006**, 43, 7727-7751.; *Smart Mater. Struct.* **2013**, 22, 104012; *Appl. Phys. Lett.*, **2016**, 108, 012903; *J. Appl. Polym. Sci.* **2016**, 133, 43258], however, further investigations on self-healing capability after breakdown have not been found. We has tried to investigate the possible self-healing of the DEA after its breakdown, but the device cannot recover.
- (2) In this study, the self-healing property of CNT/PVA hydrogel is the main concern rather than the whole DEA. It is believed that we have demonstrated comprehensively and discussed fully on self-healing property of CNT/PAV hydrogel after systematic research in this paper. Therefore, none modification are suggested to do.

Our modification to the manuscript: None.

Comment #4. *In figure 2c and 2d, the author discussed about resistance changing. It is better to show the conductivity changing rather than resistance. Then, you may find that the conductivity is almost unchanged with different strain, which would be more persuasive.*

Our response: We thank the Reviewer for the comments on our work.

- (1) We agree with the Reviewer that it is better to show the conductivity change rather than resistance. We planned to get the conductivity from sheet resistance of the hydrogel according to the following equation:

$$\sigma = \frac{1}{R_s \cdot t} \quad (1)$$

where σ , R_s , and t are the conductivity, sheet resistance, and thickness of the hydrogels, respectively.

- (2) We find it is very difficult to measure the sheet resistance practically during stretching of the hydrogel, according to the following reasons: (a) The sample needs to maintain square during measurement using four-point probe station (ST2253-F01 Suzhou JingGe Inc. Figure R3a), but it is difficult to maintain the shape during stretching; (b) It is difficult to configure the four-point probe station and the stretcher (Figure R3b) together during the measurement; (c) Water evaporation can affect the accuracy during the conductivity measurement (Figure R3c), while during the resistance measurement, the hydrogel electrode is encapsulated.

(3) If the Reviewer has a more straightforward and effective method, we would be delighted to know and push our research forward by it.

Figure R3. Photographs relevant to conductivity measuring. (a) A photograph of ST2253-F01 four-point probe station. (b) A photograph of homemade stretcher. (c) A photograph of a half-dried CNT/PVA electrode after water evaporation during stretching.

Our modification to the manuscript: None.

Comment #5 Recently, people can produce high voltage to drive DEA by using self-powered technique [Adv. Funct. Mater. 26, (2016) 4906-4913; Nano Energy 38 (2017) 91–100; Adv. Funct. Mater. (2017), 27, 1603788]. It is better the author can report some of these work in their introductions.

Our response: We thank the Reviewer for the comments on our work. We agree that it necessary to report relevant investigations on high voltage driven DEA self-powered technique.

Our modification to the manuscript:

(1) In the revised manuscript, on page 1, line 18, “Among them, dielectric elastomer actuator (DEA) has drawn considerable attentions since the use of electrical potential is an efficient way to actuate soft robots [4,8,9].” is changed into “Dielectric elastomer actuator (DEA) among them, composed of a soft dielectric polymer

sandwiched between two compliant electrodes, has drawn considerable attentions for actuating soft robots efficiently using electrical potential [4,8-12].”

(2) In the Reference section, Refs 10, 11, 12 are added.

Reviewer #2

Comments:

The authors reported carbon nanotube and polyvinyl alcohol-based dielectric elastomer actuator that has 40% more areal strain and is 200% stretchable. The new structure also showed self-healing capability due to hydrogen bonding and lower relative resistance than the other stretchable conductors. I believe the authors have presented interesting work and have addressed the previous reviewers' questions. Therefore, I would recommend the manuscript for publication after a minor revision.

Our response: We thank the Reviewer for the precious comments on our work.

Comments #1: It is not elevated that why the authors have chosen specific percentages for the chemical mixes while preparing the CNT-PVA hydrogel. i.e., 0.32 g PVA, 50g CNT, 2 hrs ultrasonication. How did they decide these numbers and times? What happens otherwise?

Our response: We thank the Reviewer for the comments on our work. We have investigated the electrical properties of hydrogels with different amount of CNTs. As shown in Figure R4, the relative resistance change of the hydrogel with 25 mg CNT is slightly higher at larger strain ($> \sim 100\%$) than that of the hydrogel with 50 mg CNT. Although the hydrogel with 75 mg CNT has better electrical conductivity than the one with 50 mg CNT, it cannot reach strain over $\sim 150\%$. It is thus reasonable to choose the sample with 50 mg CNT for the further study in the rest of the paper

Figure R4. Relative resistance changes of the hydrogel electrodes prepared using 25, 50, and 75 mg of CNTs at strain ranging from 0% to 200%.

Our modification to the manuscript:

(1) In the revised manuscript, Figure S4 is added to the Supporting Information.

- (2) In the revised manuscript, on page 5, line 7, “Figure 2b shows the current-voltage curves of the CNT/PVA hydrogel at different strain.” is changed to “Figure 2b shows the current-voltage curves of the CNT/PVA hydrogel with 50 mg CNT at different strain.”
- (3) In the revised manuscript, on page 5, line 13, “Hydrogels with different amount of CNTs are investigated to obtain the optimized electrode for the DEA. Figure S4 shows the relative resistance changes of hydrogels with different amount of CNTs. The sample with 25 mg CNT is slightly higher than that of the hydrogel with 50 mg CNT at higher strain (> ~100%). Although the hydrogel with 75 mg CNT has better electrical conductivity than the one with 50 mg CNT, it cannot reach strain over ~150%. Therefore, hydrogel with 50 mg CNT was chosen as the electrode for the following study.” is added.

Comments #2: It is stated that the hydrogel could repeat its performance up to 1800 cycles without any degradation. Did the authors repeat this lifetime test to the sample they cut after the self-healing process? Are there any different results observed?

Our response: We thank the Reviewer for the comments on our work. According to the suggestion from the Reviewer, the stability test for the hydrogel after the cutting-healing process was investigated under 20% strain for ~2000 cycles, as shown in Figure R5. The hydrogel electrode can maintain its electromechanical performance within ~500 cycles. Then the electrical signal starts drifting. After 2000 cycle, the relative resistance change is 2 times larger than the original value. The possible reason is still under investigation.

Figure R5. The stability test of the CNT/PVA hydrogel after cutting-self healing process.

Our modification to the manuscript:

- (1) In the revised manuscript, Figure S6 is added to the Supporting Information.

- (2) In the revised manuscript, on page 6, line 23, “Stability of the hydrogel after the cutting-healing test was further investigated by subjecting it to a cycling test for 2000 cycles (Figure S6). The hydrogel electrode can maintain its electromechanical performance within ~500 cycles. Then the electrical signal starts drifting. After 2000 cycle, the relative resistance change is 2 times larger than the original value. The possible reason is still under investigation.” is added.

Comments #3: *Writing can be improved.*

Our response: We thank the Reviewer for the comments on our work. We refine and improve the manuscript after reviewing it carefully.

Our modification to the manuscript:

- (1) In the revised manuscript on page 1, line 12, “In contrast to the rigid-bodied counterparts, soft robots composed of intrinsically soft and/or extensible materials with relatively large number of freedom degrees provide an opportunity to bridge the gap between machines and people, since they can deform and absorb much of the energy arising from a collision [1,4-7].” is changed to “In contrast to rigid-bodied counterparts, soft robots composed of intrinsically soft and/or extensible materials with relatively large number of freedom degrees provide an opportunity to bridge the gap between machines and human being, since they can deform and absorb much of the energy arising from a collision [1,4-7].”
- (2) In the revised manuscript on page 1, line 16, “Development of soft bodies with large deformation, high energy density, and short response time is the key challenge for creating soft robots [5].” is changed to “Development of soft bodies with large deformation, high energy density, and short response time is key challenge to fabricate soft robots [5].”
- (3) In the revised manuscript on page 1, line 18, “Among them, dielectric elastomer actuator (DEA) has drawn considerable attentions since the use of electrical potential is an efficient way to actuate soft robots [4,8-12].” is changed to “Dielectric elastomer actuator (DEA) among them, composed of a soft dielectric polymer sandwiched between two compliant electrodes, has drawn considerable attentions for actuating soft robots efficiently using electrical potential [4,8-12].”
- (4) In the revised manuscript on page 2, line 5, “DEAs consist of a soft dielectric polymer sandwiched between two compliant electrodes.” is deleted.

- (5) In the revised manuscript on page 2, line 3, “Compliant electrode is an important component for DEAs. They must be able to synchronously follow the large strains of the elastomer, but without generating an opposing stress or losing conductivity [13].” is changed to “Compliant electrode is an important component for DEAs, since they must be able to synchronously follow large strains of the elastomer without generating an opposing stress or losing conductivity [13].”
- (6) In the revised manuscript on page 2, line 7, “Up to date, a variety of materials including carbon grease [13], carbon nanotubes (CNTs) [15], nanowires [13], and graphene [16] have been investigated as the electrodes for DEAs.” is changed to “A variety of materials including carbon grease [13], carbon nanotubes (CNTs) [15], nanowires [13], and graphene [16] have been investigated as the electrodes for DEAs up to now.”
- (7) In the revised manuscript on page 2, line 11, “but it has been found that these materials have poor mechanical adhesion with DEAs, and are not applicable for miniaturization [14,27].” is changed to “but it is found that these materials have poor mechanical adhesion with DEAs, and are not applicable for miniaturization [14,27].”
- (8) In the revised manuscript on page 2, line 14, “Electrical conductivity of the composite electrodes is typically low and its stiffness is generally higher than dielectric polymers [17,28]. Furthermore, there is a trade-off between electrical conductivity and stiffness of the composite electrodes: to increase electrical conductivity of the composite electrodes by filling more active material will increase their stiffness [22, 29].” is changed to “Electrical conductivity of the composite electrodes is typically low and its stiffness is generally higher than dielectric polymers [17,28]. Furthermore, there is a trade-off between electrical conductivity and stiffness of the composite electrodes: to increase electrical conductivity of the composite electrodes by filling more active material will increase their stiffness [22, 29].”
- (9) In the revised manuscript on page 2, line 18, “Ionic conductive polymers have been investigated as potential candidates as the compliant electrodes for DEAs due to their high electrical conductivity at large deformation [17-19].” is changed to “Ionic conductive polymers have been investigated as potential candidates to compliant electrodes for DEAs due to their high electrical conductivity at large deformation [17-19].”

- (10) In the revised manuscript on page 2, line 27, “The areal strains generated by DEAs based on CNT/PVA hydrogel electrode are almost two times larger than the ones based on pure PVA hydrogel electrode. The large stretchability, high electrical conductivity, and self-healing capability of the CNT/PVA composite electrode demonstrate its potential applications in DEAs.” is changed to “Areal strain generated by DEAs based on CNT/PVA hydrogel electrode are almost two times larger than that based on pure PVA hydrogel electrode. Large stretchability, high electrical conductivity, and self-healing capability of the CNT/PVA composite electrode prospects its applications in DEAs.”
- (11) In the revised manuscript on page 3, line 14, “The dielectric elastomer VHB 4910 (3M Company Shanghai Branch) was first glued to a rigid acrylic frame with radius D for fixation. After that, two layers of CNT/PVA hydrogel with radius d ($d < D$) were attached to the top and bottom faces of the dielectric elastomer membrane. Two pieces of copper tapes were bonded to the ends of the two electrodes, respectively, for electromechanical measurements.” is changed to “Having a layer of dielectric elastomer VHB 4910 (3M Company Shanghai Branch) fixed on a rigid acrylic frame with radius D , two CNT/PVA hydrogel films with radius d ($d < D$) were attached to the top and bottom faces of the dielectric elastomer membrane, respectively. Two pieces of copper tapes aiming at electromechanical measurements were bonded to the ends of the electrodes afterwards.”
- (12) In the revised manuscript on page 4, line 1, “Voltage signal was amplified by a high voltage amplifier (TRC-2020P, Teslaman). To measure the real time displacement of the DEAs as an in-plane movement, a thin paper tape was attached on the edge of the electrode area. A laser displacement sensor (LK-G4000A, Kenyence) was used to measure the real time displacement of the tape by continuously tracking and recording the displacement of this tape. The areal strain (ϵ_{area}) of the DEAs was calculated as following: ...by the electrodes in the original and actuated state, respectively” on page 4, line 1 in the revised manuscript, is changed to “A voltage signal was amplified by a high voltage amplifier (TRC-2020P, Teslaman) to actuate the DEAs. A thin paper tape was attached on the edge of electrode area of the DEAs for measuring real time in-plane displacement. A laser displacement sensor (LK-G4000A, Kenyence) was used for tracking real time displacement of the tape

continuously. Areal strain (ϵ_{area}) of the DEAs was calculated as following: ...by the electrodes at original and actuated state, respectively.”

- (13) In the revised manuscript on page 4, line 12, “The PVA aqueous solution and the CNT solution were mixed together to form the CNT/PVA hydrogel (Figure 1b) with the introduction of sodium tetraborate solution. The infilling of CNTs improves the electrical conductivity of PVA. The conductivity of the CNT/PVA is measured to be ~ 0.71 s/cm, higher than that of pure PVA (~ 0.22 s/cm). Figure S1 shows the SEM images of CNTs used for the preparation of CNT/PVA hydrogel electrode. Figure S2 shows the SEM images of CNT/PVA electrode after a freeze-drying process, with a porous structure. Due to the high amount of PVA used, the CNTs seem to be wrapped by PVA.” is changed to “PVA aqueous solution and CNT solution were mixed together to form the CNT/PVA hydrogel (Figure 1b) by introducing sodium tetraborate solution. Infilling of CNTs seen from Figure S1 and S2 improves electrical conductivity of PVA up to ~ 0.71 s/cm higher than that of pure PVA (~ 0.22 s/cm), while the CNTs seem to be wrapped by PVA resulting from high amount of PVA.”
- (14) In the revised manuscript on page 4, line 17, “The VHB elastomer was pre-stretched to a strain $200\% \times 200\%$ in two directions and glued to a rigid acrylic frame for fixation. Then two pieces of CNT/PVA hydrogel were applied to the surfaces of the elastomer to serve as the top and bottom electrodes, respectively.” is changed to “The VHB elastomer was pre-stretched to a strain $200\% \times 200\%$ in two directions and fixed to a rigid acrylic frame. Then two pieces of CNT/PVA hydrogel were applied to the surfaces of the elastomer serving as the top and bottom electrodes, respectively.”
- (15) In the revised manuscript on page 4, line 22, “The CNT/PVA hydrogel electrode has a high failure strain close to 780% with a fracture strength of ~ 12 kPa, while VHB layer has a failure strain of $\sim 930\%$ and fracture strength of ~ 450 kPa. In addition, the stress of hydrogel electrode at all strains is less than the VHB dielectric layer, ensuring that the hydrogel electrode is compliant to the dielectric layer.” is changed to “The CNT/PVA hydrogel electrodes have high failure strain close to 780% with fracture strength of ~ 12 kPa, while VHB layer has failure strain of $\sim 930\%$ and fracture strength of ~ 450 kPa. In addition, stress of hydrogel electrodes at all strains

is less than the VHB dielectric layer, ensuring hydrogel electrodes are compliant to the dielectric layer.”

- (16) In the revised manuscript on page 5, line 7, “Figure 2b shows the current-voltage curves of the CNT/PVA hydrogel at different strains. As the applied strain increases, the slopes of the current-voltage curves decrease, indicating the increase in the resistance of the CNT/PVA hydrogel electrode. For an electrode used for DEA, high conductivity with large stretchability and good linearity are needed.” is changed to “Figure 2b shows the current-voltage curves of the CNT/PVA hydrogel at different strain. As the applied strain increases, the slope of the current-voltage curves decrease indicating the increase in the resistance of the CNT/PVA hydrogel electrodes. High conductivity with large stretchability and good linearity of the electrodes are necessary in the field of DEAs.”
- (17) In the revised manuscript on page 5, line 21, “The existed hysteresis loops can be contributed from the hysteresis of the VHB elastomer or the PVA used for hydrogel preparation.” is changed to “The existed hysteresis loops can be contributed to the hysteresis of the VHB elastomer or the PVA used for hydrogel preparation.”
- (18) In the revised manuscript on page 7, line 1, “The large stretchability, high conductivity, and self-healing capability of the CNT/PVA hydrogel electrodes enable its potential application in DEAs. Figure 4 shows a systematic study of the electromechanical performance of the DEAs based on the CNT/PVA hydrogel electrodes. The areal strain of the DEAs was measured using the experimental setup shown in Figure 4a. The detailed information can be found in the Experimental section. The thickness of electrode in the experiment is ~1 mm.” is changed to “Large stretchability, high conductivity, and self-healing capability of the CNT/PVA hydrogel electrodes enable their potential application in DEAs. Figure 4 shows a systematic study of the electromechanical performance of the DEAs based on the CNT/PVA hydrogel electrodes. Areal strain of the DEAs was measured using the experimental setup shown in Figure 4a. Detailed information can be found in the Experimental section. Thickness of the electrode in the experiment is ~1 mm.”
- (19) In the revised manuscript on page 7, line 8, “When the DEA was subject to a high voltage, the DEA reduces in thickness and expands in area due to the effect of Maxwell stress [3]. The actuation performance of the DEA based on the CNT/PVA hydrogel electrodes at different applied voltages is shown in Figure 4c. With the

increase in the applied voltage, the areal strain of the DEA based on CNT/PVA hydrogel electrode increases. At an applied voltage of ~2 kV, the areal strain is more than 40%. Compared to the DEA based on pure PVA hydrogel, the electromechanical performance of the DEAs using the CNT/PVA hydrogel electrodes is much enhanced.” is changed to “When the DEA was subjected to a high voltage, it reduces in thickness and expands in area due to the effect of Maxwell stress [3]. Actuation performance of the DEA based on the CNT/PVA hydrogel electrodes at different applied voltages is shown in Figure 4c. With the increase of applied voltage, areal strain of the DEA increases. At an applied voltage of ~2 kV, the areal strain is more than 40%. Compared with DEAs based on pure PVA hydrogel, DEAs using the CNT/PVA hydrogel electrodes is much enhanced.”

(20) In the revised manuscript on page 7, line 16, “The actuation properties of the DEAs based on different electrodes was measured at different frequencies, as shown in figure 4d. The applied voltage was maintained at ~1.5 kV. The areal strain of all the two samples reduced as the frequency of applied voltage increased. The DEA based on the CNT/PVA hydrogel electrodes provides larger areal strains at all the frequencies than the ones based on pure PVA hydrogel, suggesting a better frequency response behaviour of the actuator enabled by the CNT/PVA hydrogel electrodes.” is changed to “The actuation properties of the DEAs based on different electrodes was measured at different frequencies under ~1.5 kV, as shown in figure 4d. Areal strain of all the two samples reduced as the frequency of applied voltage increased. DEAs based on the CNT/PVA hydrogel electrodes provide larger areal strains at all the frequencies than the ones based on pure PVA hydrogel, suggesting a better frequency response behaviour of the actuator enabled by the CNT/PVA hydrogel electrodes.”

(21) In the revised manuscript on page 7, line 23, “Although the CNT/PVA hydrogel electrode not only provides high electromechanical properties, but also has the capability of self-healing, it is notable that the performance of the electrode can become degraded due to the loss of water. The effect of water loss on the actuation property of the DEA based on CNT/PVA hydrogel electrodes was studied, as shown in Figure 5.” is changed to “Although the CNT/PVA hydrogel electrodes not only provide high electromechanical properties, but also have the capability of self-healing, it is notable that electric performance of the electrode can become degraded

due to the loss of water. Effect of water loss on the actuation property of the DEAs based on CNT/PVA hydrogel electrodes was studied, as shown in Figure 5.”

List of Changes

- (2) In the revised manuscript, Figure 1e is replaced by a higher resolution image, as shown in Figure R1.
- (3) In the revised manuscript, Figure 4c and 4d and their figure captions are replaced.
- (4) In the revised manuscript, on page 7, line 13, “Areal strain of the DEA based on VHB 4905 (blue curve in Figure 4c) is higher than that based on VHB 4910 (black curve in Figure 4c), while the former breaks down at 1.8 kV. The lower breakdown voltage of the DEA based on VHB 4905 can be contributed to lower thickness during expansion.” is added.
- (5) In the revised manuscript, on page 7, line 21, “The DEA based on CNT/PVA hydrogel with VHB 4905 (blue curve in Figure 4d) has 2 times larger areal strain than that of the device based on VHB 4910 (blue curve in Figure 4d).” is added.
- (6) In the revised manuscript, on page 1, line 18, “Among them, dielectric elastomer actuator (DEA) has drawn considerable attentions since the use of electrical potential is an efficient way to actuate soft robots [4,8,9].” is changed into “Dielectric elastomer actuator (DEA) among them, composed of a soft dielectric polymer sandwiched between two compliant electrodes, has drawn considerable attentions for actuating soft robots efficiently using electrical potential [4,8-12].”
- (7) In the Reference section, Refs 10, 11, 12 are added.
- (8) In the revised manuscript, Figure S4 is added to the Supporting Information.
- (9) In the revised manuscript, on page 5, line 7, “Figure 2b shows the current-voltage curves of the CNT/PVA hydrogel at different strain.” is changed to “Figure 2b shows the current-voltage curves of the CNT/PVA hydrogel with 50 mg CNT at different strain.”
- (10) In the revised manuscript, on page 5, line 13, “Hydrogels with different amount of CNTs are investigated to obtain the optimized electrode for the DEA. Figure S4 shows the relative resistance changes of hydrogels with different amount of CNTs. The sample with 25 mg CNT is slightly higher than that of the hydrogel with 50 mg CNT at higher strain ($> \sim 100\%$). Although the hydrogel with 75 mg CNT has better electrical conductivity than the one with 50 mg CNT, it cannot reach strain over $\sim 150\%$. Therefore, hydrogel with 50 mg CNT was chosen as the electrode for the following study.” is added.
- (11) In the revised manuscript, Figure S6 is added to the Supporting Information.

- (12) In the revised manuscript, on page 6, line 23, “Stability of the hydrogel after the cutting-healing test was further investigated by subjecting it to a cycling test for 2000 cycles (Figure S6). The hydrogel electrode can maintain its electromechanical performance within ~500 cycles. Then the electrical signal starts drifting. After 2000 cycle, the relative resistance change is 2 times larger than the original value. The possible reason is still under investigation.” is added.
- (13) In the revised manuscript on page 1, line 12, “In contrast to the rigid-bodied counterparts, soft robots composed of intrinsically soft and/or extensible materials with relatively large number of freedom degrees provide an opportunity to bridge the gap between machines and people, since they can deform and absorb much of the energy arising from a collision [1,4-7].” is changed to “In contrast to rigid-bodied counterparts, soft robots composed of intrinsically soft and/or extensible materials with relatively large number of freedom degrees provide an opportunity to bridge the gap between machines and human being, since they can deform and absorb much of the energy arising from a collision [1,4-7].”
- (14) In the revised manuscript on page 1, line 16, “Development of soft bodies with large deformation, high energy density, and short response time is the key challenge for creating soft robots [5].” is changed to “Development of soft bodies with large deformation, high energy density, and short response time is key challenge to fabricate soft robots [5].”
- (15) In the revised manuscript on page 1, line 18, “Among them, dielectric elastomer actuator (DEA) has drawn considerable attentions since the use of electrical potential is an efficient way to actuate soft robots [4,8-12].” is changed to “Dielectric elastomer actuator (DEA) among them, composed of a soft dielectric polymer sandwiched between two compliant electrodes, has drawn considerable attentions for actuating soft robots efficiently using electrical potential [4,8-12].”
- (16) In the revised manuscript on page 2, line 5, “DEAs consist of a soft dielectric polymer sandwiched between two compliant electrodes.” is deleted.
- (17) In the revised manuscript on page 2, line 3, “Compliant electrode is an important component for DEAs. They must be able to synchronously follow the large strains of the elastomer, but without generating an opposing stress or losing conductivity [13].” is changed to “Compliant electrode is an important component for DEAs, since they

must be able to synchronously follow large strains of the elastomer without generating an opposing stress or losing conductivity [13].”

- (18) In the revised manuscript on page 2, line 7, “Up to date, a variety of materials including carbon grease [13], carbon nanotubes (CNTs) [15], nanowires [13], and graphene [16] have been investigated as the electrodes for DEAs.” is changed to “A variety of materials including carbon grease [13], carbon nanotubes (CNTs) [15], nanowires [13], and graphene [16] have been investigated as the electrodes for DEAs up to now.”
- (19) In the revised manuscript on page 2, line 11, “but it has been found that these materials have poor mechanical adhesion with DEAs, and are not applicable for miniaturization [14,27].” is changed to “but it is found that these materials have poor mechanical adhesion with DEAs, and are not applicable for miniaturization [14,27].”
- (20) In the revised manuscript on page 2, line 14, “Electrical conductivity of the composite electrodes is typically low and its stiffness is generally higher than dielectric polymers [17,28]. Furthermore, there is a trade-off between electrical conductivity and stiffness of the composite electrodes: to increase electrical conductivity of the composite electrodes by filling more active material will increase their stiffness [22, 29].” is changed to “Electrical conductivity of the composite electrodes is typically low and its stiffness is generally higher than dielectric polymers [17,28]. Furthermore, there is a trade-off between electrical conductivity and stiffness of the composite electrodes: to increase electrical conductivity of the composite electrodes by filling more active material will increase their stiffness [22, 29].”
- (21) In the revised manuscript on page 2, line 18, “Ionic conductive polymers have been investigated as potential candidates as the compliant electrodes for DEAs due to their high electrical conductivity at large deformation [17-19].” is changed to “Ionic conductive polymers have been investigated as potential candidates to compliant electrodes for DEAs due to their high electrical conductivity at large deformation [17-19].”
- (22) In the revised manuscript on page 2, line 27, “The areal strains generated by DEAs based on CNT/PVA hydrogel electrode are almost two times larger than the ones based on pure PVA hydrogel electrode. The large stretchability, high electrical conductivity, and self-healing capability of the CNT/PVA composite electrode

demonstrate its potential applications in DEAs.” is changed to “Areal strain generated by DEAs based on CNT/PVA hydrogel electrode are almost two times larger than that based on pure PVA hydrogel electrode. Large stretchability, high electrical conductivity, and self-healing capability of the CNT/PVA composite electrode prospects its applications in DEAs.”

(23) In the revised manuscript on page 3, line 14, “The dielectric elastomer VHB 4910 (3M Company Shanghai Branch) was first glued to a rigid acrylic frame with radius D for fixation. After that, two layers of CNT/PVA hydrogel with radius d ($d < D$) were attached to the top and bottom faces of the dielectric elastomer membrane. Two pieces of copper tapes were bonded to the ends of the two electrodes, respectively, for electromechanical measurements.” is changed to “Having a layer of dielectric elastomer VHB 4910 (3M Company Shanghai Branch) fixed on a rigid acrylic frame with radius D , two CNT/PVA hydrogel films with radius d ($d < D$) were attached to the top and bottom faces of the dielectric elastomer membrane, respectively. Two pieces of copper tapes aiming at electromechanical measurements were bonded to the ends of the electrodes afterwards.”

(24) In the revised manuscript on page 4, line 1, “Voltage signal was amplified by a high voltage amplifier (TRC-2020P, Teslaman). To measure the real time displacement of the DEAs as an in-plane movement, a thin paper tape was attached on the edge of the electrode area. A laser displacement sensor (LK-G4000A, Kenyence) was used to measure the real time displacement of the tape by continuously tracking and recording the displacement of this tape. The areal strain (ϵ_{area}) of the DEAs was calculated as following: ...by the electrodes in the original and actuated state, respectively” on page 4, line 1 in the revised manuscript, is changed to “A voltage signal was amplified by a high voltage amplifier (TRC-2020P, Teslaman) to actuate the DEAs. A thin paper tape was attached on the edge of electrode area of the DEAs for measuring real time in-plane displacement. A laser displacement sensor (LK-G4000A, Kenyence) was used for tracking real time displacement of the tape continuously. Areal strain (ϵ_{area}) of the DEAs was calculated as following: ...by the electrodes at original and actuated state, respectively.”

(25) In the revised manuscript on page 4, line 12, “The PVA aqueous solution and the CNT solution were mixed together to form the CNT/PVA hydrogel (Figure 1b) with the introduction of sodium tetraborate solution. The infilling of CNTs improves the

electrical conductivity of PVA. The conductivity of the CNT/PVA is measured to be ~ 0.71 s/cm, higher than that of pure PVA (~ 0.22 s/cm). Figure S1 shows the SEM images of CNTs used for the preparation of CNT/PVA hydrogel electrode. Figure S2 shows the SEM images of CNT/PVA electrode after a freeze-drying process, with a porous structure. Due to the high amount of PVA used, the CNTs seem to be wrapped by PVA.” is changed to “PVA aqueous solution and CNT solution were mixed together to form the CNT/PVA hydrogel (Figure 1b) by introducing sodium tetraborate solution. Infilling of CNTs seen from Figure S1 and S2 improves electrical conductivity of PVA up to ~ 0.71 s/cm higher than that of pure PVA (~ 0.22 s/cm), while the CNTs seem to be wrapped by PVA resulting from high amount of PVA.”

(26) In the revised manuscript on page 4, line 17, “The VHB elastomer was pre-stretched to a strain $200\% \times 200\%$ in two directions and glued to a rigid acrylic frame for fixation. Then two pieces of CNT/PVA hydrogel were applied to the surfaces of the elastomer to serve as the top and bottom electrodes, respectively.” is changed to “The VHB elastomer was pre-stretched to a strain $200\% \times 200\%$ in two directions and fixed to a rigid acrylic frame. Then two pieces of CNT/PVA hydrogel were applied to the surfaces of the elastomer serving as the top and bottom electrodes, respectively.”

(27) In the revised manuscript on page 4, line 22, “The CNT/PVA hydrogel electrode has a high failure strain close to 780% with a fracture strength of ~ 12 kPa, while VHB layer has a failure strain of $\sim 930\%$ and fracture strength of ~ 450 kPa. In addition, the stress of hydrogel electrode at all strains is less than the VHB dielectric layer, ensuring that the hydrogel electrode is compliant to the dielectric layer.” is changed to “The CNT/PVA hydrogel electrodes have high failure strain close to 780% with fracture strength of ~ 12 kPa, while VHB layer has failure strain of $\sim 930\%$ and fracture strength of ~ 450 kPa. In addition, stress of hydrogel electrodes at all strains is less than the VHB dielectric layer, ensuring hydrogel electrodes are compliant to the dielectric layer.”

(28) In the revised manuscript on page 5, line 7, “Figure 2b shows the current-voltage curves of the CNT/PVA hydrogel at different strains. As the applied strain increases, the slopes of the current-voltage curves decrease, indicating the increase in the resistance of the CNT/PVA hydrogel electrode. For an electrode used for DEA, high

conductivity with large stretchability and good linearity are needed.” is changed to “Figure 2b shows the current-voltage curves of the CNT/PVA hydrogel at different strain. As the applied strain increases, the slope of the current-voltage curves decrease indicating the increase in the resistance of the CNT/PVA hydrogel electrodes. High conductivity with large stretchability and good linearity of the electrodes are necessary in the field of DEAs.”

- (29) In the revised manuscript on page 5, line 21, “The existed hysteresis loops can be contributed from the hysteresis of the VHB elastomer or the PVA used for hydrogel preparation.” is changed to “The existed hysteresis loops can be contributed to the hysteresis of the VHB elastomer or the PVA used for hydrogel preparation.”
- (30) In the revised manuscript on page 7, line 1, “The large stretchability, high conductivity, and self-healing capability of the CNT/PVA hydrogel electrodes enable its potential application in DEAs. Figure 4 shows a systematic study of the electromechanical performance of the DEAs based on the CNT/PVA hydrogel electrodes. The areal strain of the DEAs was measured using the experimental setup shown in Figure 4a. The detailed information can be found in the Experimental section. The thickness of electrode in the experiment is ~1 mm.” is changed to “Large stretchability, high conductivity, and self-healing capability of the CNT/PVA hydrogel electrodes enable their potential application in DEAs. Figure 4 shows a systematic study of the electromechanical performance of the DEAs based on the CNT/PVA hydrogel electrodes. Areal strain of the DEAs was measured using the experimental setup shown in Figure 4a. Detailed information can be found in the Experimental section. Thickness of the electrode in the experiment is ~1 mm.”
- (31) In the revised manuscript on page 7, line 8, “When the DEA was subject to a high voltage, the DEA reduces in thickness and expands in area due to the effect of Maxwell stress [3]. The actuation performance of the DEA based on the CNT/PVA hydrogel electrodes at different applied voltages is shown in Figure 4c. With the increase in the applied voltage, the areal strain of the DEA based on CNT/PVA hydrogel electrode increases. At an applied voltage of ~2 kV, the areal strain is more than 40%. Compared to the DEA based on pure PVA hydrogel, the electromechanical performance of the DEAs using the CNT/PVA hydrogel electrodes is much enhanced.” is changed to “When the DEA was subjected to a high voltage, it reduces in thickness and expands in area due to the effect of Maxwell

stress [3]. Actuation performance of the DEA based on the CNT/PVA hydrogel electrodes at different applied voltages is shown in Figure 4c. With the increase of applied voltage, areal strain of the DEA increases. At an applied voltage of ~2 kV, the areal strain is more than 40%. Compared with DEAs based on pure PVA hydrogel, DEAs using the CNT/PVA hydrogel electrodes is much enhanced.”

(32) In the revised manuscript on page 7, line 16, “The actuation properties of the DEAs based on different electrodes was measured at different frequencies, as shown in figure 4d. The applied voltage was maintained at ~1.5 kV. The areal strain of all the two samples reduced as the frequency of applied voltage increased. The DEA based on the CNT/PVA hydrogel electrodes provides larger areal strains at all the frequencies than the ones based on pure PVA hydrogel, suggesting a better frequency response behaviour of the actuator enabled by the CNT/PVA hydrogel electrodes.” is changed to “The actuation properties of the DEAs based on different electrodes was measured at different frequencies under ~1.5 kV, as shown in figure 4d. Areal strain of all the two samples reduced as the frequency of applied voltage increased. DEAs based on the CNT/PVA hydrogel electrodes provide larger areal strains at all the frequencies than the ones based on pure PVA hydrogel, suggesting a better frequency response behaviour of the actuator enabled by the CNT/PVA hydrogel electrodes.”

(33) In the revised manuscript on page 7, line 23, “Although the CNT/PVA hydrogel electrode not only provides high electromechanical properties, but also has the capability of self-healing, it is notable that the performance of the electrode can become degraded due to the loss of water. The effect of water loss on the actuation property of the DEA based on CNT/PVA hydrogel electrodes was studied, as shown in Figure 5.” is changed to “Although the CNT/PVA hydrogel electrodes not only provide high electromechanical properties, but also have the capability of self-healing, it is notable that electric performance of the electrode can become degraded due to the loss of water. Effect of water loss on the actuation property of the DEAs based on CNT/PVA hydrogel electrodes was studied, as shown in Figure 5.”